# Study protocol: quantitative fibronectin to help decision-making in women with symptoms of preterm labour (QUIDS) part 2, UK Prospective Cohort Study

Sarah Jane Stock,[1,2] Lisa M Wotherspoon,[1] Kathleen Anne Boyd,[3] Rachel K Morris,[4] Jon Dorling,[5] Lesley Jackson,[6] Manju Chandiramani,[7] Anna L David,[8,9] Asma Khalil,[10] Andrew Shennan,[11] Victoria Hodgetts Morton,[4] Tina Lavender,[12] Khalid Khan,[13] Susan Harper-Clarke,[14] Ben Mol,[15] Richard D Riley,[16] John Norrie,[17] Jane Norman[1]

For numbered affiliations see end of article.

**Correspondence to**
Dr Sarah Jane Stock;
sarah.stock@ed.ac.uk

## ABSTRACT

**Introduction** The aim of the QUIDS study is to develop a decision support tool for the management of women with symptoms and signs of preterm labour, based on a validated prognostic model using quantitative fetal fibronectin (fFN) concentration, in combination with clinical risk factors.

**Methods and analysis** The study will evaluate the Rapid fFN 10Q System (Hologic, Marlborough, Massachusetts, USA) which quantifies fFN in a vaginal swab. In QUIDS part 2, we will perform a prospective cohort study in at least eight UK consultant-led maternity units, in women with symptoms of preterm labour at $22^{+0}$ to $34^{+6}$ weeks gestation to externally validate a prognostic model developed in QUIDS part 1. The effects of quantitative fFN on anxiety will be assessed, and acceptability of the test and prognostic model will be evaluated in a subgroup of women and clinicians (n=30). The sample size is 1600 women (with estimated 96–192 events of preterm delivery within 7 days of testing). Clinicians will be informed of the qualitative fFN result (positive/negative) but be blinded to quantitative fFN result. Research midwives will collect outcome data from the maternal and neonatal clinical records. The final validated prognostic model will be presented as a mobile or web-based application.

**Ethics and dissemination** The study is funded by the National Institute of Healthcare Research Health Technology Assessment (HTA 14/32/01). It has been approved by the West of Scotland Research Ethics Committee (16/WS/0068).

**Version** Protocol V.2, Date 1 November 2016.

**Trial registration number** ISRCTN 41598423andCPMS: 31277.

## INTRODUCTION

The overall aim of the QUIDS study is to develop a decision support tool for the management of women with symptoms and signs of preterm labour, based on a validated prognostic model using quantitative fetal

### Strengths and limitations of this study

► Validation of a prognostic model in a separate prospective cohort study.
► Health economic analysis to determine cost-effectiveness from National Health Service perspective.
► Not a randomised control trial to test effectiveness of the model on improved patient outcomes.

fibronectin (fFN) testing. The study has been conceptually divided into two parts. In this, the protocol for QUIDS part 2, we detail the protocol for a prospective cohort study. This will externally validate a prognostic model developed in QUIDS part 1.[1] More detailed background about the diagnosis of preterm labour and background to the study is provided in the introduction of QUIDS Protocol part 1.[1]

fFN is a biochemical test of preterm labour which has the potential to help improve diagnosis of impending preterm delivery.[2] Much of the evidence about fFN to date relates to the qualitative fFN test, which provides a positive or negative result on the basis of a single threshold of $50\,\text{ng/mL}$.[2 3] This test has been largely replaced with the Rapid fFN 10Q System, which provides a concentration of fFN (quantitative fFN), and as a continuous variable, may be a more useful predictor of preterm delivery. fFN is now only available with a quantitative analyser in the UK, but there is no consensus as to which women to use the test in, or how to interpret the results.

The QUIDS study will address this evidence gap by providing evidence about the potential value of the quantitative fFN test, along with guidance about how to interpret results. Here,

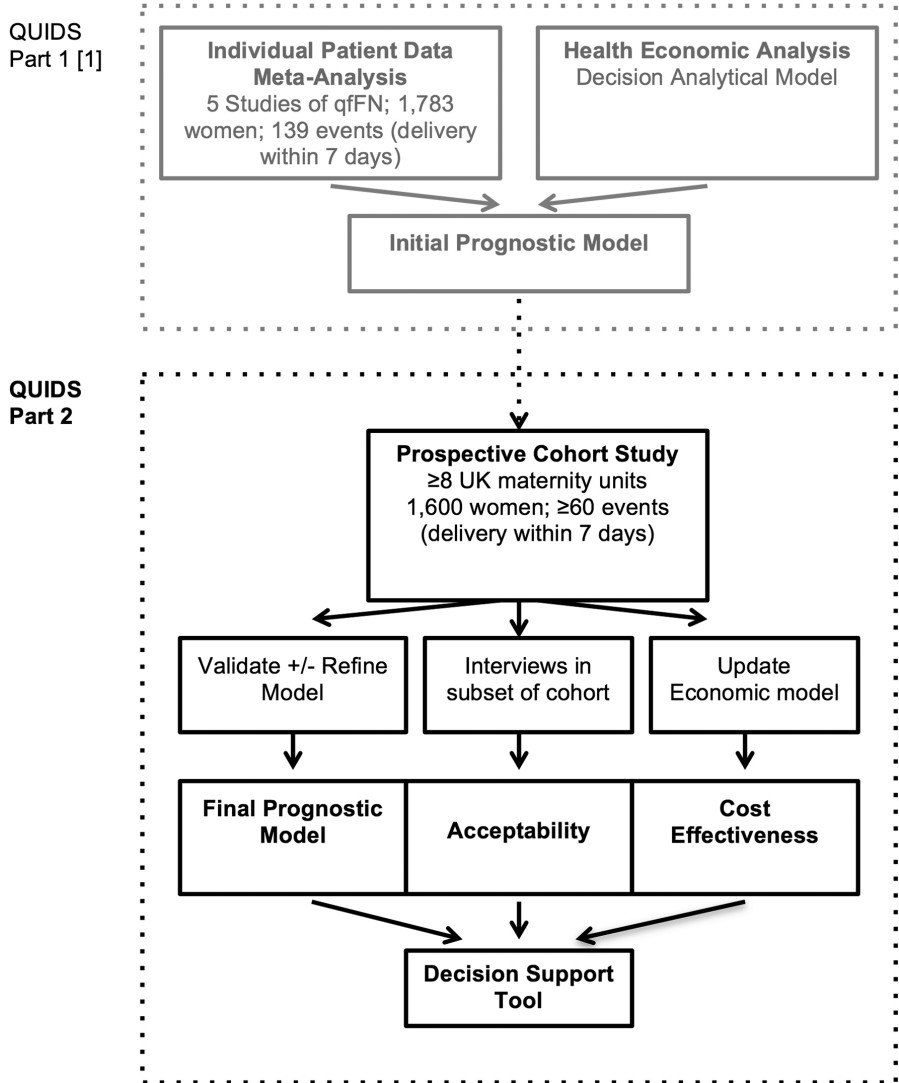

**Figure 1** Flow chart illustrating the design of QUIDS study and conceptual division into part 1 and part 2. qfFN, quantitative fetal fibronectin.

we detail the protocol for external validation of a prognostic model developed in QUIDS part 1.[1]

## METHODS AND ANALYSIS
### Aims and methodologies

The aim of the QUIDS study is to develop a decision support tool for the management of women with symptoms and signs of preterm labour, based on a validated prognostic model using quantitative fFN testing.

The study protocol has been divided into two parts (see flow chart figure 1). The protocols for parts 1 and 2 are reported in separate manuscripts.

In QUIDS Protocol part 1, we have described how we will perform (1) an individual participant data (IPD) meta-analysis and (2) and economic analysis. The protocol details how we will develop and internally validate a prognostic model using quantitative fFN (as a continuous variable) and other risk (prognostic) factors and to evaluate the added value of quantitative

fFN towards this prognostic model performance. We will also provide an economic rationale for the prognostic model and analyse its cost-effectiveness from the perspective of the National Health Service (NHS).

In this, the QUIDS Protocol part 2, we will detail the prospective cohort study to externally validate and, if necessary, refine the prognostic model. This will be performed in at least eight UK hospitals with different settings (rural/urban) and different levels of neonatal care facilities. In addition, acceptability of quantitative fFN testing, and effects on maternal anxiety will be performed. We will assess the potential cost-effectiveness of the final prognostic model/decision support tool. This additional analysis will allow us to model the full costs and effect impacts of the different prognostic model and compare these in a cost-effectiveness analysis to provide an evidence-based economic rationale for implementing the diagnostic tool in the NHS.

## Endpoints

The primary endpoint of the prognostic model is spontaneous preterm delivery within 7 days of qfFN test, in women less than 36 weeks' gestation. This was influenced by the preceding QUIDS Qualitative study, which included focus group consultation to determine the decisional needs of women, their partners and clinicians (online supplementary material). It is also a recognised clinically important endpoint, as antenatal steroids (which significantly reduce morbidity and mortality in preterm babies)[4] are most effective if delivery occurs within 7 days of administration.

A secondary endpoint suggested by QUIDS Qualitative study (online supplementary material) consultation, was delivery within 48 hours of qfFN test. This analysis will be performed if feasible to do so within the constraints of the data available for model development and validation.[1]

## Health technologies being assessed

The trial will evaluate the Rapid fFN 10Q System (Hologic, Marlborough, Massachusetts, USA). This provides a concentration of fFN (ng/mL or INVALID) in a vaginal swab sample in 10 min. It is now the only commercially available fFN test system, and replaces the TLiQ rapid analyser system, which provided a qualitative fFN result (POSITIVE or NEGATIVE) based on a threshold of 50 ng/mL. The Rapid fFN 10Q system is a point of care test, which clinical staff can easily perform. All reagents for fFN testing can be stored at room temperature and specimen collection kits, reagents, cassettes and the 10Q analyser can be kept in clinical areas where women with symptoms of preterm labour are assessed so they can be conveniently accessed.

Vaginal swab samples are analysed by lateral flow; solid-phase immunochromatographic assay (the Rapid fFN Cassette) and interpreted in the 10Q Rapid analyser. Two hundred microlitres of the sample is pipetted into the sample application well of the Rapid fFN Cassette using a polypropylene or polyethylene pipette. The sample will then flow from an absorbent pad across a nitrocellulose membrane via capillary action through a reaction zone containing murine monoclonal anti-fFN antibody conjugated to blue microspheres (conjugate). The conjugate, embedded in the membrane, will be mobilised by the flow of the sample. The sample will then flow through a zone containing goat polyclonal anti-human fibronectin antibody that captures the fibronectin-conjugate complexes. The remaining sample will flow through a zone containing goat polyclonal antimouse IgG antibody that captures unbound conjugate, resulting in a control line. After 10 min of reaction time, the intensities of the test line and control line are interpreted with the 10Q Rapid analyser and a printed result provided as a concentration in ng/mL (0≥500 ng/mL) or INVALID. The result is invalid if the test does not meet internal quality controls (QC) that are performed automatically with every test. In the event of an invalid result, the test can be repeated with any remaining clinical specimen.

A QC can be performed by a reusable Rapid fFN 10Q QCette QC Device, which verifies that the analyser performance is within specification.

## Target population

The target population is pregnant women attending hospital with signs and symptoms of preterm labour.

## Validation and refinement of prognostic model
### Population

The prospective cohort study will include women with signs and symptoms of preterm labour at $22^{+0}$ to $34^{+6}$ weeks gestation in whom admission, transfer or treatment is being considered. These will be recruited from at least eight sites with a mix of rural/urban settings, and have different levels of neonatal care facilities, over 12 months.

### Eligibility criteria

The following inclusion criteria will apply at screening assessment (all apply):
► Women who are $22^{+0}$ to $34^{+6}$ weeks (or earlier gestation if the fetus is considered potentially viable).
► Women showing signs and symptoms of preterm labour which may include any or all of back pain, abdominal cramping, abdominal pain, light vaginal bleeding, vaginal pressure, uterine tightenings or contractions.
► Women where hospital admission, interhospital transfer or treatment (antenatal steroids, tocolysis or magnesium sulfate) are being considered due to signs of preterm labour.
► Women aged 16 years or above.

The broad inclusion criteria reflect current clinical practice and enable the generalisability of the results of the trial for routine clinical care. We will include women who reattend 7 days or more after initial recruitment with signs and symptoms of preterm labour and also women who remain symptomatic but undelivered 7 days later in whom repeat testing by the clinician is deemed to be appropriate. This will be in line with manufacturer's recommendation for fFN testing.

The following inclusion criteria will apply on speculum examination:
► Cervical dilation ≤3 cm.
► Intact membranes.
► No significant vaginal bleeding, as judged by the clinician.
► Once it has been established that the women meet the above criteria, on speculum examination, the fFN swab can be taken.

Participants who sign the consent but are not eligible on examination to have an fFN swab taken will still be enrolled and have outcome data collected.

The following exclusion criteria will apply:
► Contraindication to vaginal examination (eg, placenta praevia).
► Higher order multiple pregnancy (triplets or more).
► Moderate or severe vaginal bleeding.

► Cervical dilatation greater than 3 cm.

► Confirmed rupture of membranes.

► Sexual intercourse, vaginal examination or transvaginal ultrasound in the preceding 24 hours factors may invalidate results. These women will be initially excluded from the study, but can be included if still symptomatic after 24 hours, when fFN accuracy will be restored.

### Co-enrolment

This trial involves validating a decision support tool relating to a test that is currently commonly used in clinical practice. As such, there are no additional interventions. Co-enrolment in other non-interventional trials will be allowed. Co-enrolment in trials of tocolytic treatments or other management strategies that may influence timing of delivery as a primary outcome will not be allowed. Participation in QUIDs would not preclude babies being subsequently involved in interventional trials. Co-enrolment will be recorded in the electronic case report form (CRF).

### Setting

The prospective cohort study will take place in at least eight consultant-led obstetric units in the UK. More than 93% of pregnant women in the UK deliver in consultant-led units.[5 6] The vast majority of women with symptoms of preterm labour will present to a consultant-led unit for assessment, either directly or following advice from their community midwife or general practitioner.

The study will not include any community maternity units (staffed by midwives, with or without involvement of non-obstetric medical staff), which cover a small proportion of women, mainly in remote and rural areas. In the Perinatal Collaborative Transport Study of perinatal transfers in Scotland,[7] which involved 52 727 births, only 69 (0.13%) women were transferred to a consultant-led obstetric unit from community maternity units, and only a proportion of these were for suspected preterm labour. The small number of women cared for in community maternity units means their inclusion would not be an efficient use of study resources.

Given that management of women with symptoms of preterm labour and interhospital transfer patterns are likely to vary depending on level of available neonatal care and distance to transfer, we will include a mixture of hospitals with different levels of neonatal care facilities in both rural and urban settings. We will include units with special care units (providing special care for their own local population), local neonatal units (providing special care and high dependency care and a restricted volume of intensive care) and neonatal intensive care units (larger intensive care units providing the whole range of medical, and sometimes surgical neonatal care for their local population and for babies and their families referred from the neonatal network in which they are based, and other networks when necessary). The hospitals will be chosen from different geographical settings (rural/urban) and from different regions of the UK.

If additional units wish to participate in the study, we will consider including them, to increase recruitment rates. The UK Reproductive Health and Childbirth specialty group (clinical study group) have contributed to the study protocol and support the proposed trial.

### Participant selection and enrolment

Women with signs and symptoms of preterm labour will be identified on presentation to obstetric services. A member of clinical staff, usually the doctor or midwife assessing the woman, will identify potentially eligible participants, provide a participant information leaflet and invite consent. A suitably trained member of clinical staff (doctor or midwife) or research team will consent participants.

Posters and leaflets will be situated in antenatal areas of participating hospitals to alert women that the study is taking place, and women will be allowed as much time as possible to consider participation without unduly delaying further clinical assessment. Participants will receive adequate oral and written information and appropriate participant information and informed consent forms will be provided.

### Screening for eligibility

The clinical likelihood of preterm delivery is usually evaluated by history and examination, which includes abdominal palpation, to assess strength and frequency of uterine contractions. If preterm labour is suspected, a vaginal speculum examination is performed where the cervix is inspected for dilatation, and evidence of vaginal bleeding and membrane rupture assessed. Swabs for fFN are usually taken at this point. Potential participants in the QUIDS study will be identified after the initial assessment and provided with information about the study. A combined 'Screening and Consent Form' will be used as a self-screening tool for potentially eligible participants. Informed consent will take place before speculum examination and the fFN swab has been taken. This approach means that samples are collected at routine speculum examination, as they would be if fFN is implemented in clinical practice, and participants avoid an additional vaginal examination.

### Ineligible and non-recruited participants

Certain exclusion criteria can only be assessed at speculum examination (eg, vaginal bleeding or evidence of ruptured membranes), so a proportion of women will not be eligible for fFN testing after consent is given. These women will still be enrolled and delivery outcomes collected. The decision whether to use this data for analysis will be the decision of the chief investigator and statisticians.

### Withdrawal of study participants

Women will be able to withdraw consent for us of their data at any time until the end of the study.

**Table 1** QUIDS study assessments

| Visit | Attendance with signs and symptoms preterm labour | | | |
| --- | --- | --- | --- | --- |
| | Screening and recruitment | 24–48 hours | 1–6 months | Delivery |
| Inclusion/exclusion criteria | ◉ | | | |
| Participant information sheet | ◉ | | | |
| Consent form | ◉ | | | |
| Demographics | ◉ | | | |
| Obstetric history | ◉ | | | |
| Symptoms and signs | ◉ | | | |
| Quantitative fetal fibronectin (concentration ng/mL) | ◉ | | | |
| Cervical length scan (if available) | ◉ | | | |
| State Trait Anxiety Inventory Questionnaire | ◉ | ◉ | | |
| Delivery details | | | | ◉ |
| Neonatal outcomes | | | | ◉ |
| Qualitative Acceptability Questionnaires (subgroup n=30) | | | ◉ | |

## Study assessments (see table 1)

### Eligibility assessment (screening and recruitment)

Women presenting with signs and symptoms of preterm labour will be identified on presentation to obstetric services. The doctor or midwife assessing the woman will identify potentially eligible participants and provide an invitation letter and short information leaflet.

After the woman has had the opportunity to consider whether she would like to participate, she will be asked to complete the Screening and Consent Form. The clinician will then decide whether the fFN test can be carried out. If the test can be carried out (according to manufacturer's guidelines), then the participant will be fully enrolled and that their delivery outcomes will still be collected.

If the woman declines to participate and she is willing to provide a reason for this, the reason given will be entered on to an anonymous log. Baseline demographics will be collected on consenting women, together with height and weight, information on medical history, obstetric history, estimated date of delivery and presenting signs and symptoms.

The original consent form will be stored in the Investigator Site File, a copy is given to the woman, a copy added to the medical notes and a copy sent to the Trial Office.

After providing consent, the participant will be asked to complete a short State Trait Anxiety Inventory (STAI) questionnaire and complete a contact details form. They will also be issued with a letter thanking them for taking part in the trial and giving details of the second questionnaire to be completed.

### Sample collection

Samples for analysis will be taken with an fFN specimen collection kit, which consists of a sterile polyester tipped swab and a specimen transport tube containing 1 mL extraction buffer (an aqueous solution containing protease inhibitors and protein preservatives including aprotinin, bovine serum albumin and sodium azide). During speculum examination the sterile swab will be lightly rotated across the posterior fornix of the vagina for 10 s to absorb vaginal secretions. Samples should be taken before any other swabs (eg, for microbiology) or cervical manipulation and the speculum lubricated with normal saline as other lubricants may interfere with the antibody-antigen reaction of the test. Following specimen collection the swab should be removed, immersed in extraction buffer, the shaft of the swab snapped off and the transport tube sealed.

Before analysis, samples are gently mixed and as much liquid as possible expressed from the swab by rolling the tip against the inside of the tube.

### Initial fFN test

The sample taken will be run at a near bedside Hologic Rapid fFN 10Q analyser, specially adapted for the QUIDS study. As fFN (or other similar biochemical tests of preterm labour) are part of standard care, it would be unethical to blind clinicians from the qualitative fFN result. The analyser will thus reveal a qualitative fFN result (positive/negative/invalid based on a 50 ng/mL threshold) for clinicians to base clinical decision-making on, according to local protocols. The quantitative fFN result, however, will be stored as a three-letter code, blinding caregivers from the result. Samples will be run as per manufacturers instructions (described above in the section 'health technologies being assessed').

### Repeat fFN tests

If there is clinical indication for further fFN tests (eg, because of ongoing symptoms of preterm labour after 7 days), the results will also be recorded.

### Labour/delivery/neonatal assessments

Admission for delivery will not be a formal study visit, but data will be collected using information recorded in the participant's notes. Delivery data will be collected on the maternal outcomes of delivery, including method of delivery, indication for delivery method, onset of labour, date and gestation of delivery and blood loss.

### Questionnaires

All participants who are eligible to participate will be asked to complete a STAI questionnaire before the speculum examination. The same questionnaire will be repeated on 24–48 hours postexamination. The second questionnaire will be provided on paper with a prepaid envelope to be returned by post to the Trial Office. If not returned by post, the Trial Office may try to contact the participant (with the contact details provided), to complete the questionnaire over the phone.

### Safety and quality assessments

The Hologic Rapid fFN 10Q analyser has integrated QC measures, and we will keep records of these as well as any additional staff training that occurs after the study starts. It is recommended that a daily precalibrated reusable QC cassette be inserted and analysed every 24 hours to verify that the analyser performance is within specification. A daily QC should be performed if one has not been done in the preceding 24 hours before a patient test is to be done. Logs of results are stored on the machine and can be downloaded, and we will also ask the participating sites to keep a monthly paper log of QC tests done. Each patient test has an internal QC, with a procedural control line that verifies the threshold level of signal by the instrument. Sample flow detection ensures the sample travels across the cassette properly, and confirms absence of conjugate aggregation. We believe that these measures will help ensure the validity of results. However, to provide further evidence of integrity and comparability of results from each site, we will request that all participating sites enrol in the Wales External Quality Assurance Scheme (WEQAS) Point of Care Quality Assurance Scheme. WEQAS will provide a sample for analysis to each site bimonthly, and provide reports on analyser performance and variability.[8]

### Data collection

#### Data for prognostic model validation and update of health economic model

We will collect data on all of the candidate predictors considered for inclusion in the prognostic model developed in the IPD meta-analysis (quantitative fFN concentration, previous spontaneous preterm labour, gestation at fFN test, age, ethnicity, body mass index, smoking, deprivation index, number of uterine contractions in set time period, cervical dilatation, vaginal bleeding, previous cervical treatment for cervical intraepithelial neoplasia, cervical length (measured by transvaginal cervical length; when available), singleton/multiple pregnancy, tocolysis and fetal sex). Outcome data will include gestational age at delivery, date and time of delivery, administration of treatments for preterm labour (steroids, antibiotics, tocolysis, magnesium sulfate) duration hospital admission, hospital transfer, onset of labour (preterm prelabour rupture of membranes, idiopathic preterm birth, medically indicated preterm birth (and indication)), place of delivery (base hospital, other hospital, outwith hospital), mode of delivery, neonatal admission, neonatal complications, perinatal mortality, congenital anomaly, sex and birth weight.

Screening data and data about quantitative fFN testing will be collected on paper-based CRFs and research midwives will input these into the web-based electronic database. Clinical outcome data will be collected from the medical records.

### Maternal acceptability and anxiety

Maternal anxiety will be measured pretest and post-test (24–48 hours) using the validated STAI questionnaire. Acceptability of fFN testing and the decision support will be assessed using follow-up interviews (face to face or telephone, according to maternal preference) which will be conducted with a subgroup of participants (n=30) purposively sampled and stratified according to geographical location, outcome (preterm labour or not) and anxiety scores. Acceptability will also be assessed in a cohort of clinicians (n=30).

### Statistics and sample size calculation

Guidance for external validation suggests at least 10 events (preterm delivery within 7 days of test) are required for each covariate included in a prognostic model.[9 10] Data from the cohorts included in our IPD meta-analysis suggest an event rate of between 6% and 12%.[1] Based on these estimates a sample size of 1600 will provide 96 and 192 events (preterm delivery within 7 days).

A UK study has shown that 8.9% of pregnant women present with symptoms of preterm labour and are eligible for quantitative fFN,[11] and we anticipate 50% recruitment rate is achievable, thus, overall 4.5% of maternities could be recruited. We will initially include eight units in the cohort study with a combined delivery rate of approximately 36 000 per annum. We anticipate that we will achieve target recruitment within 12 months (1 year x 36 000 x 0.089 x 0.5=1602). If, however, the recruitment rate or event rate is lower than predicted, we will increase the number of sites included in the study and/ or the recruitment period, to ensure that a minimum of 60 events (preterm delivery within 7 days of test) are achieved, allowing for external validation of at least six covariates in our model.

It is possible that the IPD meta-analysis will find there is potential added value of combining quantitative fFN

testing with cervical length measurement.[12 13] As cervical length measurement has significant resource requirement (estimated NHS cost £68.16 per test) and lack of out-of-hours provision further limits availability in many NHS hospitals, we think it is very unlikely that cervical length scanning will improve performance of the prognostic model to such a degree as to make it cost-effective. We will assess the incremental costs and effects of cervical length measurement in the proposed health economic model performed in parallel with the IPD meta-analysis, and will feed into design considerations during the first iteration of the prognostic model.

If inclusion of cervical length ultrasound is found to be potentially cost-effective, we will assess the feasibility of including it in the prospective cohort study. We anticipate that including cervical length measurement in the prospective cohort study would be extremely difficult in the current NHS setting as the majority of units do not have 24 hours availability of transvaginal ultrasound and/or trained personnel to perform scans. Inclusion of cervical length would also likely decrease recruitment rate (due to need for additional transvaginal ultrasound examination) and require significant additional resources.

## Analysis
### Validation of prognostic model
The prognostic model developed in the IPD will be externally validated using data collected in the prospective cohort data, using the measures of discrimination and calibration described in QUIDS Protocol part 1,[1] including $R^2$, C statistic, calibration slope, calibration in the large and calibration plots of observed versus predicted risks across deciles (with Loess smoother). The average performance of the model will be summarised across the centres in the cohort study. Between-centre heterogeneity in performance will also be summarised, and reduced (if necessary) by recalibration techniques regarding the strategy for the choice of baseline risk (intercept). That is, the predictor effects will not be modified from the IPD meta-analysis model, but the intercept may need to be tailored to improve validation in UK centres (eg, for rural settings). Based on the findings, a final model and its implementation strategy will then be recommended for use.

### Economic analysis
The economic model will be refined, integrated and updated with data from the prospective study cohort, so as the most up to date and validated evidence is used to inform a cost-effectiveness decision. Such an iterative approach to economic evaluation is now well established.[14 15] The care pathway following diagnosis will be included in the economic analysis, using data from the cohort study such as the diagnostic test accuracy data, resource use data (ie, steroid use, other medications, time in hospital, hospital transfer) and secondary outcome data (ie, treatment of side effects, morbidity, mortality) so as to capture the full costs and effect impacts (quality of life, morbidity and mortality) for both the mother and baby. Resource use data will be combined with unit cost information from the British National Formulary[16] and NHS reference costs.[17 18] Outcomes will be reported as the incremental cost per correct diagnosis, and incremental cost per quality-adjusted life year gained of the qfFN prognostic model compared with current practice (no qualitative fFN model). The analysis will adhere to the National Institute for Health and Care Excellence reference case and the recommended guidelines for decision modelling and reporting of economic analyses.[18] Probabilistic sensitivity analysis will be undertaken to explore how uncertainty in the model inputs impact on the cost-effectiveness outcome.[19]

### Acceptability of fFN testing and effects on anxiety
Maternal anxiety will be measured before and after quantitative fFN testing using the validated STAI. The STAI Form Y is a widely used tool for measuring both temporary 'state anxiety' and the more general, long-standing 'trait anxiety'. The STAI is designed for the self-reported assessment of the intensity of feelings of apprehension, tension, nervousness and worry. STAI-Anxiety scores increase in response to physical danger and psychological stress, making it highly appropriate for this study. The use of STAI in pregnancy studies is discussed by Hundley et al and we will interpret the results accordingly.[20]

The questionnaire will be administered prior to fFN testing (baseline) and 24–48 hours after the test, to assess early reactions to the test and any acute anxiety prompted by the result of the test. We will also be able to assess any differences in those presented with a high-risk or low-risk result. Although it might be interesting to assess anxiety again in the latter stages of pregnancy, it is likely that, in this population, many pregnancies will not reach full term. Thus, we believe our strategy of repeat questionnaire administration will allow measurement of longer-term anxiety induced or alleviated by the test, while minimising bias due to preterm or term delivery itself or loss to follow-up.

Follow-up interviews will be performed with a subgroup of participants (n=30) to enable deeper exploration of women's views regarding fFN testing, to gain insight into the rationale for responses given in the questionnaires. Interviews will be conducted following confirmation of pregnancy status. Acceptability of the prognostic model will also be assessed with women and a group of clinicians. All interviews will be audio recorded with consent, and field notes taken to ensure an audit trail.

### Decision support
We will develop a decision support tool in accordance with the guidelines produced by the International Patient Decision Aid Standards Collaboration.[21] Scoping of decisional requirements and how data should be presented was performed during focus group consultation as part of QUIDS Qualitative (online supplementary material). A prototype decision support tool incorporating

the initial prognostic model, developed as part of the IPD meta-analysis, will be tested with women and clinicians, as part of the acceptability studies described above. A final version will be updated with the validated (and, if necessary revised) prognostic model generated from the prospective cohort study. The multidisciplinary trial steering committee (TSC) will oversee the development process, and decide how material is selected for inclusion.

### Trial management and oversight arrangements
#### Project management group
The trial will be coordinated by a project management group (PMG), consisting of the grant holders (chief investigator and coapplicants), the trial manager, representatives from the Study Office and CHaRT (the supporting Clinical Trials Unit (CTU)), plus service user representatives (PAG). The PMG will meet approximately every 4 months by teleconference or face to face.

The trial manager based in Edinburgh will oversee the study and will be accountable to the chief investigator. The trial manager supported by the trial administrator(s) will take responsibility for the day-to-day transaction of study activities. They will be supported by the CTU at CHaRT to provide expertise and guidance. The trial manager will be responsible for checking the CRFs for completeness, plausibility and consistency. Any queries will be resolved by the investigator or delegated member of the trial team.

A Delegation Log will be prepared for each site, detailing the responsibilities of each member of staff working on the trial.

#### TSC and data monitoring committee
A combined TSC and data monitoring committee (DMC) will oversee the conduct and progress of the trial. The terms of reference of the Committee will be developed separately. Members of the TSC/DMC will consist of experts and two patient representatives.

### PATIENT AND PUBLIC INVOLVEMENT
Patient representatives were consulted during the protocol development and have been invited to join the PMG and the TSC, and will thus be involved in the recruitment to, and conduct of, the study. Coauthor SH-C is a patient representative. Prior to commencing QUIDS, we performed a qualitative study to determine the decisional needs of pregnant women with signs and symptoms of preterm labour, their partners and their caregivers. This is described in the separate protocol 'QUIDS Qualitative' (online supplementary material). The end product of QUIDS will be a decision support aid to help clinicians, women and their partners decide on management of threatened preterm labour, based on the results of the quantitative fFN. In QUIDS Qualitative women and clinicians indicated that they would prefer this to be on web-based or mobile app-based format, presenting the risk of preterm birth within 7 days of testing. Social media will be used to signpost publications and conference presentations and highlight important findings. Twitter and Facebook will be used to disseminate findings to professional organisations, charities, stakeholders and the public. Communication to the general public will further be facilitated by our close links with charities such as Tommys.[22]

### ETHICS AND DISSEMINATION
The study will be conducted in accordance with the principles of Good Clinical Practice (GCP). Local research and development approvals will be obtained prior to commencement of the study at each site.

On completion of the study, the study data will be analysed and tabulated, and a clinical study report will be prepared in accordance with GCP guidelines. Results will be communicated to the academic community via the scientific literature, attendance at conferences and invited presentations. Summaries of results will also be made available to investigators for dissemination within clinics. We anticipate that the decision support will be made available as web-based application that will be made freely available, so clinicians can access it easily and it can be readily translatable into UK practice. If it is found to be effective in ruling out preterm delivery, it is likely that it will decrease unnecessary costly, and potentially harmful treatments in women who have symptoms suggestive of preterm labour but do not deliver early .

### How patients are involved in this study
Patient representatives were consulted during the protocol development and have been invited to join the PMG and the TSC. Prior to commencing QUIDS, we performed a qualitative study to determine the decisional needs of pregnant women with signs and symptoms of preterm labour, their partners and their caregivers. This is described in the separate protocol "QUIDS Qualitative" (online supplementary material). The end product of QUIDS will be a decision support aid to help clinicians, women and their partners decide on management of threatened preterm labour, based on the results of the quantitative fFN. In QUIDS Qualitative, women and clinicians indicated that they would prefer this to be on web-based or mobile app-based format, presenting the risk of preterm birth within 7 days of testing.

**Author affiliations**
[1]Tommy's Centre for Maternal and Fetal Health, University of Edinburgh MRC Centre for Reproductive Health, Queen's Medical Research Institute, Edinburgh, UK
[2]School of Women's and Infants' Health, University of Western Australia, Crawley, Western Australia, Australia
[3]Health Economics and Health Technology Assessment, Institute of Health and Wellbeing, University of Glasgow, Glasgow, UK
[4]School of Clinical and Experimental Medicine, University of Birmingham, Birmingham, UK
[5]Neonatal Unit, Queen's Medical Centre, Nottingham, UK
[6]Neonatal Unit, Royal Hospital for Children Glasgow, Glasgow, UK
[7]Department of Surgery & Cancer, Reproductive Biology, Imperial College Healthcare NHS Trust, Queen Charlotte's and Chelsea Hospital, London, UK

[8] Institute for Women's Health, University College London Medical School, London, UK

[9] 86-96 Chenies Mews, University College London Medical School London, London, UK

[10] St. George's Medical School, University of London, London, UK

[11] Maternal and Fetal Research Unit, King's College London, London, UK

[12] School of Nursing, Midwifery and Social Work, University of Manchester, Manchester, UK

[13] Centre for Primary Care and Public Health, Queen Mary University of London, London, UK

[14] Patient and Public Involvement Representative

[15] Department of Obstetrics and Gynaecology, Monash University, Melbourne, Victoria, Australia

[16] Research Institute for Primary Care and Health Sciences, Keele University, Keele, UK

[17] Edinburgh Clinical Trials Unit, University of Edinburgh, Edinburgh, UK

**Contributors** SJS, KAB, RKM, JD, LJ, MC, ALD, AK, AS, VH-M, TL, KK, SH-C, BM, RDR, JN and JEN developed the protocol. SJS, LMW, RDR, KAB, TL and JN drafted the protocol. RKM, JD, LJ, MC, ALD, AK, AS, VH-M, KK, SH-C, BM and JEN reviewed and commented on the protocol.

**Funding** This project was funded by the National Institute of Healthcare Research Health Technology and Assessment (Reference 14/32/01). The views expressed are those of the authors and not necessarily those of the NHS, the NIHR or the Department of Health.

**Competing interests** SJS and JEN work at the University of Edinburgh, who received £1000 sponsorship from Hologic to support a meeting (The Society of Reproductive Investigation and MRC Centre for Reproductive Health Scientific Symposium on Targeting Inflammation to Improve Reproductive Health across the Lifecourse – August 2017). AS has in the past (over last 5 years; not in the last 3 years) received funding for expenses related to advisory board and internal staff education from Hologic. MC received sponsorship from Hologic to organise an educational teaching focusing on prediction of Preterm Birth at the 2017 annual meeting of the British Maternal and Fetal Medicine Society. Hologic, the makers of fFN have provided analysers and technical support for their use to sites participating in the QUIDS prospective cohort study. They have no access to the data, or other involvement in the conduct, data analysis, interpretation of results or decision to publish the results of the study.

**Patient consent** Not required.

**Ethics approval** A favourable ethical opinion has been obtained from the appropriate REC (reference 16/WS/0068).

**Provenance and peer review** The study was extensively peer reviewed as part of the process of gaining grant funding from the NIHR HTA (14/32/01).

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
