## [Reviewer comments · BMJ Open]

ARTICLE DETAILS

TITLE (PROVISIONAL)	Study Protocol: Quantitative Fibronectin to help Decision-making in women with Symptoms of Preterm Labour (QUIDS) Part Two- UK Prospective Cohort Study
AUTHORS	Stock, Sarah; Wotherspoon, Lisa; Boyd, Kathleen; Morris, R. K.; Dorling, Jon; Jackson, Lesley; Chandiramani, Manju; David, Anna; Khalil, Asma; Shennan, Andrew; Hodgetts Morton, Victoria; Lavender, Tina; Khan, Khalid; Harper-Clarke, Susan; Mol, Ben; Riley, Richard; Norrie, John; Norman, Jane

VERSION 1 – REVIEW

REVIEWER	Ibrahim A. Abdelazim Professor of Obstetrics, Gynecology and Reproductive Medicine, Ain Shams University, Cairo, Egypt. Consultant of Obstetrics, Gynecology and Reproductive Medicine, Ahmadi Hospital, KOC, Kuwait.
REVIEW RETURNED	03-Dec-2017

GENERAL COMMENTS	Dear Respectable Colleagues Good Day I am grateful to give me the opportunity to review the Study Protocol: Quantitative Fibronectin to help Decision-making in women with Symptoms of Preterm Labour (QUIDS) Part Two- Prospective Cohort Study. This study protocol is excellent, and can be published because of;  1. The major neonatal morbidities following the PTL, and to prevent the unnecessary admissions, and medications when the diagnosis of PTL is not sure. 2. The aim of this part of the QUIDS study is to develop a decision support tool for the management of women with symptoms, and signs of PTL, based on a validated prognostic model using quantitative fetal Fibronectin (fFN) concentration, in combination with clinical risk factors. 3. Nature of the study as prospective cohort study, which will include women with signs and symptoms of PTL at 22+0 to 34+6 weeks gestation, recruited from at least eight UK consultant-led maternity units, over 12 months. 4. Good sample size of 1600 women (with estimated 96-192 events of PTL within 7 days of testing). 5. Availability of the final validated prognostic model will be presented as a mobile or web-based application. Finally; this study protocol is excellent, and can be published. Best Regards Ibrahim A. Abdelazim
---

REVIEWER	Penelope Sheehan Department of O&G, RWH, University of Melbourne, Australia
REVIEW RETURNED	03-Jan-2018

GENERAL COMMENTS	A very interesting proposal for a study to investigate the clinical value of fetal Fibronectin testing in threatened preterm labour. The overall study is well-designed especially the power calculations taking into account the notoriously high rate of women who present in threatened preterm labour who don't deliver during their initial presenting episode. The study authors have not explicitly stated the cutoff value of fFN which will be considered as a positive result. The manuscript correctly states that the previously used qualitative test had a cutoff value of 50ng/ml but doesn't stipulate if this is to be the cutoff for the blinded test. With the new qualitative test, we have been using a cutoff of 200ng/ml. I could not find that the study authors had commented on the background risk profile of the patients. I find this interesting because I would have thought it entirely possible that this might modify the significance of the results. How, for example will they deal with a patient with a cerclage in place in the analysis? Will there be any stratification of results by history related risk factors or by treatments prior to the acute presentation? I'm not sure that the use of the STAI questionnaire will be able to differentiate between anxiety caused by symptoms and treatment (such as interhospital transfer) and anxiety related to the test result. Perhaps the addition of another pregnancy specific anxiety test such as PRAQ might be able to distinguish the two? The sentence at the end of page 13 unfortunately ends with a preposition (with).
--

VERSION 1 – AUTHOR RESPONSE

Response to Reviewers

Reviewer 1

-Many thanks to the reviewer for his kind words. No changes to text made

Reviewer 2

1. A very interesting proposal for a study to investigate the clinical value of fetal Fibronectin testing in threatened preterm labour. The overall study is well-designed especially the power calculations taking into account the notoriously high rate of women who present in threatened preterm labour who don't deliver during their initial presenting episode.

- Many thanks.

2. The study authors have not explicitly stated the cutoff value of fFN which will be considered as a positive result. The manuscript correctly states that the previously used qualitative test had a cutoff

value of 50ng/ml but doesn't stipulate if this is to be the cutoff for the blinded test. With the new qualitative test, we have been using a cutoff of 200ng/ml.

- Many thanks. Quantitative fFN (fFN concentration in ng/ml) is a continuous variable, just like age or BMI. We have chosen to analyse continuous variables as such, to avoid the arbitrary choice of cut-points that stems from categorisation and dichotomisation. Our prognostic model will thus include the concentration of quantitative fFN. To ensure this is clear, the following additions have been made to the text

- Page 5. Line 16 "This test has been largely replaced with the Rapid fFN 10Q System, which provides a concentration of fFN (quantitative fFN), and, ^as a continuous variable^, may be a more useful predictor of preterm delivery."

- Page 6 line 11 "^(as a continuous variable)^"

- Table 1 page 16 "Quantitative fFN ^(concentration ng/ml)^"

3. I could not find that the study authors had commented on the background risk profile of the patients. I find this interesting because I would have thought it entirely possible that this might modify the significance of the results. How, for example will they deal with a patient with a cerclage in place in the analysis? Will there be any stratification of results by history related risk factors or by treatments prior to the acute presentation?

- Many thanks. We agree that a single prognostic factor (such as qFFN) rarely predicts individual outcome risk accurately, and that prognostic models are needed, which utilize multiple prognostic factors in combination to improve individual risk prediction accuracy and to better discriminate the underlying risk across individuals.

Our aim is to develop a prognostic model in part one of the study; and validate/refine it in part two of the study. Candidate predictors (co-variables) for inclusion in the model, are detailed in part one of the protocol (submitted for publication alongside this; and also included in supplementary material – see page 13) . We have now additionally listed these on page 13 of this part of the protocol

... "We will collect data on all of the candidate predictors considered for inclusion in the prognostic model developed in the IPD meta-analysis ^(quantitative fFN concentration, previous spontaneous preterm labour, gestation at fFN test, age, ethnicity, BMI, smoking, deprivation index, number of uterine contractions in set time period, cervical dilatation, vaginal bleeding, previous cervical treatment for cervical intraepithelial neoplasia, cervical length [measured by transvaginal cervical length; when available], singleton/multiple pregnancy, tocolysis and fetal sex)^."

4. I'm not sure that the use of the STAI questionnaire will be able to differentiate between anxiety caused by symptoms and treatment (such as interhospital transfer) and anxiety related to the test result. Perhaps the addition of another pregnancy specific anxiety test such as PRAQ might be able to distinguish the two?

- Thank you. We agree this is important. In order to address this all participants will be asked to complete an STAI before the speculum examination (ie before test result) to provide a baseline result and asked to repeat it 24-48 hours later (ie after test result) to allow us to help discriminate between anxiety related to pregnancy and symptoms; and those related to fFN testing/qualitative fFN result. In addition, the acceptability of fFN testing will be assessed using follow up interviews (face to face or telephone, according to maternal preference) conducted in a sub-group of participants purposively sampled and stratified according to anxiety scores and outcome (preterm labour or not)– allowing

more in depth analysis of the cause of anxiety and acceptability. These strategies have been validated and successfully by co-applicants in previous studies (eg Norman, JE et al, Lancet Volume 387, No. 10033, p2106–2116, 21 May 2016). As such we do not think the PRAQ will add sufficient additional information to justify the increased questionnaire burden on participants.

5. The sentence at the end of page 13 Vaginal progesterone prophylaxis for preterm birth (the OPPTIMUM study): a multicentre, randomised, double-blind trial unfortunately ends with a preposition (with).

-Thank you .This has been amended and now reads

“.....medical history, obstetric history, estimated date of delivery and presenting signs and symptoms. “

VERSION 2 – REVIEW

REVIEWER	Penelope Sheehan university of melbourne, australia
REVIEW RETURNED	11-Feb-2018
GENERAL COMMENTS	Thanks. I still haven't got an answer to my first question. In this part of the protocol "The sample taken will be run at a near bedside Hologic Rapid fFN 10Q analyser, specially adapted for the QUIDS study." What will this bedside analyser report as positive, over 50ng/ml like the old test or a different cutoff? I think this information should be included in this paragraph (page 15). Every other query has been answered.

VERSION 2 – AUTHOR RESPONSE

We have made the following amendments in response to the comments:

- Please add the study location to the title as this is the preferred format of the journal.

We have added UK to the title so it reads Study Protocol: Quantitative Fibronectin to help Decision-making in women with Symptoms of Preterm Labour (QUIDS) Part Two- UK Prospective Cohort Study

- Please add an ethics and dissemination section to the main text of the manuscript.

There was already a section ethics and dissemination included in the manuscript, which included the subsections -Trial management and oversight arrangements; Good Clinical Practice; Dissemination . For clarity we have rearranged the headings so it stands as a separate section with no subheadings beneath it.

-Reviewer 2: I still haven't got an answer to my first question. In this part of the protocol "The sample taken will be run at a near bedside Hologic Rapid fFN 10Q analyser, specially adapted for the QUIDS study." What will this bedside analyser report as positive, over 50ng/ml like the old test or a different cutoff? I think this information should be included in this paragraph (page 15).

- we have amended the paragraph on page 15 to include the words "based on a 50ng/ml threshold" so it now reads

"The sample taken will be run at a near bedside Hologic Rapid fFN 10Q analyser, specially adapted for the QUIDS study. As fFN (or other similar biochemical tests of preterm labour) are part of standard care, it would be unethical to blind clinicians from the qualitative fFN result. The analyser will thus reveal a qualitative fFN result (positive/negative/invalid based on a 50ng/ml threshold) for clinicians to base clinical decision-making on, according to local protocols. The quantitative fFN result however, will be stored as a three-letter code, blinding caregivers from the result. Samples will be run as per manufacturers instructions (described above in the section "Health technologies being assessed"). "

In response to the latest correspondence (email 27/2/18) we have included a section "PATIENT AND PUBLIC INVOLVEMENT" under "methods and analysis". This includes some of the information already included in the section "HOW PATIENTS ARE INVOLVED IN THIS STUDY" (Under strengths and limitations).